# From Genes to Therapy: Pituitary Adenomas in the Era of Precision Medicine

**DOI:** 10.3390/biomedicines12010023

**Published:** 2023-12-21

**Authors:** Corneliu Toader, Nicolaie Dobrin, Catalina-Ioana Tataru, Razvan-Adrian Covache-Busuioc, Bogdan-Gabriel Bratu, Luca Andrei Glavan, Horia Petre Costin, Antonio Daniel Corlatescu, David-Ioan Dumitrascu, Alexandru Vlad Ciurea

**Affiliations:** 1Department of Neurosurgery, “Carol Davila” University of Medicine and Pharmacy, 020021 Bucharest, Romania; corneliu.toader@umfcd.ro (C.T.); razvan-adrian.covache-busuioc0720@stud.umfcd.ro (R.-A.C.-B.); bogdan.bratu@stud.umfcd.ro (B.-G.B.); luca-andrei.glavan0720@stud.umfcd.ro (L.A.G.); horia-petre.costin0720@stud.umfcd.ro (H.P.C.); david-ioan.dumitrascu0720@stud.umfcd.ro (D.-I.D.); prof.avciurea@gmail.com (A.V.C.); 2Department of Vascular Neurosurgery, National Institute of Neurology and Neurovascular Diseases, 077160 Bucharest, Romania; 3Neurosurgical Clinic, “Prof. Dr. N. Oblu” Emergency Clinical Hospital, 700309 Iași, Romania; 4Department of Ophthalmology, “Carol Davila” University of Medicine and Pharmacy, 020021 Bucharest, Romania; 5Department of Ophthalmology, Clinical Hospital of Ophthalmological Emergencies, 010464 Bucharest, Romania; 6Neurosurgery Department, Sanador Clinical Hospital, 010991 Bucharest, Romania

**Keywords:** pituitary adenomas, molecular mechanisms, targeted therapies, immunotherapy, biomarkers, epigenetics, transcriptomics, tumor microenvironment, mTOR inhibitors, VEGF inhibitors, tyrosine kinase, cancer progression, radiological imaging, genetic stability, hormone hypersecretion

## Abstract

This review presents a comprehensive analysis of pituitary adenomas, a type of brain tumor with diverse behaviors and complexities. We cover various treatment approaches, including surgery, radiotherapy, chemotherapy, and their integration with newer treatments. Key to the discussion is the role of biomarkers in oncology for risk assessment, diagnosis, prognosis, and the monitoring of pituitary adenomas. We highlight advances in genomic, epigenomic, and transcriptomic analyses and their contributions to understanding the pathogenesis and molecular pathology of these tumors. Special attention is given to the molecular mechanisms, including the impact of epigenetic factors like histone modifications, DNA methylation, and transcriptomic changes on different subtypes of pituitary adenomas. The importance of the tumor immune microenvironment in tumor behavior and treatment response is thoroughly analyzed. We highlight potential breakthroughs and innovations for a more effective management and treatment of pituitary adenomas, while shedding light on the ongoing need for research and development in this field to translate scientific knowledge into clinical advancements, aiming to improve patient outcomes.

## 1. Introduction

### 1.1. Definition and Categorization of Intracranial Tumors

Intracranial neoplasms, also known as brain tumors, are defined by an aberrant tissue accumulation where cell proliferation proceeds unchecked, deviating from normal cellular growth regulation mechanisms [1]. The classification of these tumor cells is complex due to their intrinsic heterogeneity. Within the realm of machine learning for visual analysis and brain tumor identification, convolutional neural networks (CNNs) have been widely adopted. A study implemented a CNN strategy using an amended EfficientNet architecture, enhanced with dense and dropout layers, to categorize 3260 T1-weighted contrast-enhanced magnetic resonance images of the brain. This approach applied min-max normalization for categorizing the images into glioma, meningioma, pituitary tumors, and non-tumorous conditions [2].

Traditionally, central nervous system (CNS) tumor classification has depended on histological evaluation, complemented by various tissue-based examinations, such as immunohistochemical and ultrastructural tests [3]. Recently, molecular biomarkers have gained prominence, offering essential diagnostic insights. Accordingly, the fifth edition of the World Health Organization (WHO) classification of CNS tumors has incorporated numerous molecular changes of clinical and pathological importance, thus improving the precision of CNS neoplasm categorization [4].

The complexity in detecting brain tumors is augmented by the variability in their location, structure, and size. The efficacy of magnetic resonance imaging (MRI) in brain tumor detection is also a subject which we will discuss later on [5]. It explores several techniques, including computational intelligence and statistical image processing methods, for the identification of brain cancer and tumors [6].

Pituitary tumors, typically benign, are linked to significant morbidity due to their strategic location, increasing size, and/or abnormal pituitary hormone secretion. Their local expansive impact can lead to complications such as headaches, visual disturbances, and cranial nerve dysfunction. Moreover, the interference with the pituitary stalk can obstruct the flow of hypothalamic hormones, causing altered hormone production. Additionally, the compression of normal pituitary tissue by the enlarging tumor may result in pituitary failure [7].

### 1.2. A Brief Overview of the Unchanged Prognosis over the Past Two Decades

The Brain Tumor Reporting and Data System (BT-RADS) has been proposed as a structured reporting methodology for post-treatment brain MRIs. BT-RADS scores provide predictive insights for likely outcomes in subsequent MRIs, indicating whether there is improvement, stability, or deterioration. This system is also beneficial in predicting clinical outcomes and overall prognosis [8].

The incurability of most intrinsic brain tumors is attributed to a combination of tumor-specific and patient-specific factors. Critical prognostic determinants in glioma patients include age at diagnosis, functional status, and histological grade. Median survival times vary widely among gliomas, from just over a year for WHO grade 4 glioblastoma to more than a decade for WHO grade 2 oligodendroglioma. The relevance of molecular markers in enhancing prognostic classifications is increasingly acknowledged; for instance, the deletion of chromosomes 1p/19q is considered a positive prognostic factor in oligodendrogliomas [9].

A reported 5.5% overall excess mortality rate in patients with non-malignant CNS tumors highlights an improvement in survival rates over the past decade. However, this continued adverse effect on survival underscores the importance of the systematic registration of these tumors [10].

## 2. Pituitary Adenomas

### 2.1. Introduction

Pituitary adenomas originate from the anterior pituitary gland and are generally slow-growing, benign neoplasms. They are classified based on size or cellular origin. The effective management of pituitary adenomas requires a multidisciplinary approach, incorporating both endocrinology and neurosurgery to achieve the best outcomes [11].

The updated classification system distinctly separates tumors of the anterior lobe (adenohypophyseal) from those of the posterior lobe (neurohypophyseal) and hypothalamic tumors, also including other neoplasms originating in the sellar region. Tumors of the anterior lobe are classified into (a) well-differentiated adenohypophyseal tumors, now referred to as pituitary neuroendocrine tumors (PitNETs, previously pituitary adenomas), (b) pituitary blastomas, and (c) two types of craniopharyngioma. This revised WHO classification emphasizes the histological subtyping of a PitNET, considering factors like tumor cell lineage, cell type, and associated characteristics. Routine immunohistochemistry for pituitary transcription factors (PIT1, TPIT, SF1, GATA3, and ERα) is crucial in this classification. The reclassification of all PitNETs/pituitary adenomas from a behavior code of “0” for benign tumors to “3” for primary malignant tumors is a subject of debate among experts [12,13].

Given the pituitary gland’s central position within the skull, pituitary tumors can exert pressure on vital brain structures as they enlarge. One common effect is the compression of optic nerves, leading to gradual vision loss, often beginning with peripheral vision. Furthermore, some pituitary adenomas may overproduce one or more hormones, resulting in hormonal imbalances that affect body functions, even when the tumors are small. Importantly, pituitary adenomas can occur in individuals of any age [14].

### 2.2. Distinction from Other Intracranial Tumors

Most pituitary adenomas appear to develop sporadically, with their continued growth being relatively rare. Currently, there are no molecular markers available that can dependably predict the behavior of these adenomas. The co-occurrence of pituitary adenomas and malignancies in the same individual could be coincidental or may suggest a shared genetic predisposition contributing to tumorigenesis. This theory is supported by a number of studies that have reported detailed cancer/tumor histories across the first, second, and third generations of family members on both parental sides [15].

About half of pituitary adenomas are known to secrete specific hormones, most frequently prolactin, growth hormone, or adrenocorticotropic hormone. Despite being histologically benign, these tumors can cause significant endocrine disturbances, leading to considerable morbidity and potentially shortening lifespan. Due to their pathophysiological endocrine secretion and proximity to critical neural and vascular structures, hormone-secreting pituitary adenomas require comprehensive management. The objectives of these management strategies include reducing the tumor’s mass effect and achieving biochemical remission [16].

## 3. Epigenetics of Pituitary Adenomas

### 3.1. Introduction to the Role of Epigenetics in Tumor Development—Specific Changes Associated with Pituitary Adenomas

Epigenetic mechanisms are crucial in mammalian development and maintaining tissue-specific gene expression. Disruptions in these processes can lead to changes in gene functionality and malignant cellular transformation. Global epigenetic alterations are now recognized as a significant hallmark of cancer. Cancer, traditionally viewed as a genetic disorder, is increasingly understood to involve both genetic mutations and epigenetic anomalies [17]. Epigenetic phenomena such as nucleosome remodeling through histone modifications, DNA methylation, and microRNA (miRNA)-mediated gene targeting play critical roles in regulating biochemical pathways vital to tumorigenesis. Moreover, mutations in genes that regulate epigenetics have strengthened the connection between epigenetics and cancer [18].

A specific study focused on the effect of DNA methylation on tumor suppressor genes (TSGs) in pituitary neuroendocrine tumors, observing that DNA methylation selectively affects TSG expression and the growth and invasiveness of PitNETs. This research also noted that all silent tumors operated upon were macroadenomas, while all functioning tumors were microadenomas, suggesting that different PitNET subtypes should be considered distinct entities [19]. Another study analyzed DNA methylation and transcriptomic profiles in non-functioning pituitary adenomas (NFPAs) and normal pituitary tissues, revealing that 10% of differentially methylated CpG sites corresponded with changes in gene expression, impacting genes involved in various tumorigenesis-related pathways [20].

Histone modifications in pituitary tumors have been linked to increased p53 expression and longer progression-free survival. Sirtuins, a class of proteins, have a higher expression in growth hormone-expressing adenomas than in nonfunctional ones and show an inverse correlation with tumor size in somatotrophs. The elevation of citrullinating enzymes has been proposed as an early marker in the pathogenesis of prolactinomas [21]. Epigenetic therapies targeting DNA methyltransferases (DNMTs) and histone deacetylases (HDACs) have been suggested to reactivate the expression of epigenetically silenced genes, potentially enhancing tumor cell sensitivity to conventional treatments like chemotherapy and radiotherapy [22].

miRNAs, small non-coding single-stranded RNAs that regulate gene expression post-transcriptionally, have been increasingly implicated in pituitary tumorigenesis. Variations in miRNA expression are linked with the development of pituitary tumors. Depending on the context, these miRNAs can act as oncosuppressors or oncogenes, underscoring their pivotal role in the onset and progression of these tumors [23,24,25].

### 3.2. Potential Therapeutic Avenues Targeting Epigenetic Modifications

The role of the histone deacetylase (HDAC) family, especially HDAC2 and HDAC3 from Class I, in the development of pituitary tumors is not fully understood. Studies have shown that HDAC2 and 3 are more highly expressed in clinically non-functioning pituitary adenomas compared to normal pituitary tissues, as evidenced by RT-PCR and immunohistochemical staining (IHC) analyses. A significant finding was observed in a study where a human NFPA-derived folliculostellate cell line, PDFS, was treated with the HDAC3 inhibitor RGFP966 for 96 h, resulting in a marked 70% reduction in cell proliferation [26].

In a different research stream, LaPierre et al. explored the role of miR-7a2 in lactotrophic cell development and prolactin hormone synthesis, following preliminary data suggesting a connection between miR-7a2 deficiency and reduced prolactin expression. Contrary to initial expectations, they found that miR-7a2 knockout led to an early increase in lactotroph cell proliferation and prolactin production during embryonic development, but this was followed by a decrease in hormone production in adulthood. In contrast, the overexpression of miR-7a2 was linked to delayed lactotroph development. Further experiments in various mouse pituitary cell types and rat prolactinoma cells showed that miR-7a2 exerts its effects by inhibiting its target gene Raf1, a known promoter of prolactin production. This finding underscores the complex regulation of prolactin production. However, more research is necessary to fully elucidate miR-7a2’s role in hyperprolactinemia and evaluate its potential as a therapeutic target [23].

## 4. Transcriptomic Insights into Pituitary Adenomas

### 4.1. Overview of Transcriptomics and Its Significance in Cancer Biology

Transcriptome profiling has emerged as a crucial tool in oncology over recent decades, providing significant prognostic and predictive insights for cancer management. This approach has transformed cancer perception from a primarily histopathological and organ-centric perspective to a more molecularly based classification, enabling more personalized diagnostics and treatment strategies.

The development of single-cell transcriptomic sequencing represents a major leap in this area, offering a detailed analysis of tumor ecosystems and deepening the understanding of tumorigenesis at the cellular level [27,28,29]. Single-cell RNA sequencing (scRNA-seq) has been particularly influential in cancer cell research, uncovering new facets of cancer biology, including the cancer stem-cell concept, treatment resistance mechanisms, and the dynamics of cancer metastasis [29,30]. Moreover, RNA sequencing and gene expression analysis are being increasingly incorporated into clinical trials to identify biomarkers predictive of immunotherapy responses. Despite these advances, challenges persist in the complexity of analysis, reproducibility and variability issues, and the interpretation of results, necessitating further study and refinement [31,32].

Analyses of altered transcriptional profiles in pituitary adenomas have revealed significant differences beyond genes involved in somatic copy number alterations (SCNAs). Tumors with disrupted genomes show more variability in gene expression compared to quieter tumors. This suggests that the processes leading to unstable genomes also result in diverse transcriptomes [33].

In prolactin-producing pituitary adenomas (PRL-PAs), research has identified widespread genomic copy number amplifications, which correlate with transcriptomic changes in this tumor subtype. High copy number variations (CNVs) in PRL-PA are associated with increased prolactin production, drug resistance, and proliferation, potentially through key genes like BCAT1. This research provides insights into the effects of genomic CNVs on transcriptomes and clinical outcomes in PRL-PA, highlighting potential therapeutic targets [34].

Pituitary adenomas have been categorized into three molecular clusters based on the transcription factor guiding their terminal differentiation. The first cluster, driven by NR5A1, includes clinically non-functioning pituitary adenomas, encompassing gonadotrophinomas and null cell adenomas. The second cluster, influenced by TBX19, contains clinically evident ACTH adenomas and silent corticotroph adenomas. The third cluster, driven by POU1F1, includes TSH-, PRL-, and GH-adenomas [35].

Differences in somatic mutations and methylation levels of genes related to cell proliferation/differentiation, miRNA/LncRNA post-transcriptional regulators, and targets of antineoplastic therapies have been observed between non-aggressive and aggressive pituitary adenomas/PitNETs. Intriguingly, the methylation profile also varies based on gender. Combining genetic-epigenetic analysis with clinical, radiological, and pathological data could prove key in predicting the behavior of pituitary adenomas/PitNETs [36].

### 4.2. Implications for Targeted Therapy Based on Transcriptomic Data

Recent research has highlighted the importance of understanding the multiomic profiles of pituitary neuroendocrine tumors (PitNETs) to develop molecularly targeted therapies, especially for those resistant to current treatments. This aligns with the broader trend of applying precision medicine in cancer treatment [37].

In corticotrophinomas, a significant finding was the discovery of a somatic mutational hotspot in the ubiquitin-specific peptidase 8 (USP8) gene in nearly half of these tumors. This gene produces a protein that impedes the downregulation of the epidermal growth factor receptor (EGFR), leading to its prolonged activation. EGFR, crucial for corticotroph function, is highly expressed in Cushing’s pituitary tumors and stimulates ACTH synthesis. The mutation in corticotrophinomas increases USP8 activity, preventing EGFR degradation and maintaining its stimulatory effect [38,39].

In acromegaly treatment, the drug pasireotide has demonstrated greater efficacy than first-generation somatostatin analogs like octreotide or lanreotide, suggesting that it could become a new standard for patients who do not respond well to first-generation analogs [40].

For prolactinomas, the expression of DRD2 and NGFR has been associated with the response to dopamine agonists (DAs). In contrast, the expressions of PTTG, ERB, and ERA do not seem to correlate with DA responsiveness or tumor aggressiveness. The responsiveness of prolactinomas to DAs varies significantly, from highly responsive to resistant [41].

Comparative studies in gonadotroph tumors have yielded new insights. Analyses comparing tumor cells to normal cells revealed that differentially expressed genes in gonadotroph tumors were mostly downregulated, unlike in somatotroph and lactotroph tumors, where they were primarily upregulated. Novel tumor-related genes like AMIGO2, ZFP36, BTG1, and DLG5 have been identified. Tumors expressing multiple hormone genes showed minimal transcriptomic heterogeneity [42].

## 5. Immunological Aspects of Pituitary Adenomas

### 5.1. Introduction to the Immune Response in Intracranial Tumors

An investigation into the immunological landscape of pituitary adenomas, using gene expression data, revealed a dominance of specific immune cell types. The most prevalent infiltrating immune cells in these tumors are M2 macrophages, followed by resting CD4+ memory T cells and mast cells. Silent pituitary tumors particularly show a higher fraction of M2 macrophages compared to other subtypes. Conversely, Cushing’s pituitary tumors, including overt and subclinical cases, have a higher proportion of CD8+ T cells than growth hormone (GH) tumors, prolactinomas, hyperthyroid tumors, and silent tumors [43].

The tumor microenvironment immune composition (TMIC) in pituitary adenomas consists of myeloid cells like tumor-associated macrophages, dendritic cells, and lymphocytes including T cells and B cells (Figure 1). The interplay between these infiltrating immune cells, their secretions, the tumor, and its host is intricate, and it can lead to either tumor-promoting or anti-tumor effects [44].

### 5.2. Opportunities and Challenges for Immunotherapy in Pituitary Adenomas

The tumor immune microenvironment (TIM) plays a crucial role in cancer prognosis and the effectiveness of immunotherapies [45]. In pituitary adenomas, T cells have been identified as the predominant immune cells across various subtypes. A relationship between tumor size, patient age, and the density of tumor-infiltrating immune cells (TIICs) has been established. Particularly in corticotroph adenomas with USP8 mutations, a significant impact on the distribution of TIICs has been noted. PAs have been classified into three immune clusters based on TIIC distributions, each with distinct immune checkpoint molecule (ICM) expression patterns, correlating with different tumor evolution and progression pathways. Cluster 1 exhibits increased CTLA4/CD86 expression, while cluster 2 is marked by elevated programmed cell death protein 1/programmed cell death 1 ligand 2 (PD1/PD-L2) expression [45].

The expression of HSPB1 is significantly higher in most tumors compared to normal tissues. High levels of HSPB1 have been associated with reduced overall survival, indicating its role in immune system regulation within various cancers. Compounds like DB11638, DB06094, and DB12695 are being investigated as HSPB1 inhibitors, underscoring its potential as a therapeutic target in invasive pituitary adenoma. Thus, HSPB1 could serve as an important biomarker, influencing tumor progression through its impact on the immune system [46].

In cancer therapy, immune checkpoint blocking has become a prominent method to enhance antitumor immunity. These checkpoints are essential for regulating immune responses in peripheral tissues and maintaining immune equilibrium (Table 1). Tumors exploit immune checkpoints and their ligands to suppress T cell activity, a strategy that allows them to evade immune destruction [47].

## 6. The Tumor Microenvironment in Pituitary Adenomas

### 6.1. Exploring the Microenvironment’s Contribution to Pituitary Adenoma Progression

In the tumor microenvironment, the presence and activity of non-tumoral cells significantly affect tumor proliferation, invasiveness, and angiogenesis.

Fibroblasts, usually in a dormant mesenchymal state, can become activated in response to various stimuli, contributing to both normal physiological processes like wound healing and pathological processes such as tumor progression. Tumor-associated fibroblasts (TAFs) are abnormally activated due to growth factors and cytokines produced by cancer cells [52]. These activated TAFs secrete a variety of molecules that are key in remodeling the extracellular matrix and facilitating tumor growth, invasiveness, metastasis, and resistance to treatment [53,54].

Interleukin-6 (IL-6) plays a role in hormone release, tumor growth and proliferation, and the production of angiogenic factors, particularly vascular endothelial growth factor-A (VEGF-A) [55]. It has been found that IL-6 stimulates the growth of GH3 rat pituitary tumor cells while inhibiting the growth of normal rat pituitary cells. IL-6 can reach the pituitary through systemic circulation and is also produced within the pituitary, having paracrine effects. In the normal adenohypophysis, folliculo-stellate (FS) cells are the primary, or possibly the sole, source of IL-6, with regulation by TNF-α. The precise role of FS cells in producing IL-6 is still under investigation [56].

A study involving 40 cases each of invasive pituitary adenoma (IPA) and non-invasive pituitary adenoma (NIPA) used the streptavidin peroxidase immunohistochemical method to examine TNF-α and IL-6 expression. The study found that both TNF-α and IL-6 were more highly expressed in IPA tissues than in NIPA, highlighting their significant roles in IPA development and progression [57].

Furthermore, the vascular endothelial growth factor (VEGF)/VEGF receptor (VEGFR) signaling pathway is closely associated with the tumor immune microenvironment. Elevated levels of VEGF-A and VEGFR1, along with a related immunosuppressive microenvironment, were observed in non-functioning PitNETs with cavernous sinus (CS) invasion. These observations point to the potential for new targeted therapies in such cases [58].

### 6.2. Strategies to Target and Modulate the Tumor Microenvironment for Therapeutic Advantage

The enhanced understanding of the immune tumor microenvironment (TME) has been pivotal in the development and clinical implementation of immuno-checkpoint inhibitors (ICIs), significantly transforming the treatment of various cancers in the past decade [59]. In light of these developments, ICIs are being explored as a new therapeutic option for aggressive pituitary adenomas/PitNETs and in the rarer instances of pituitary carcinomas [44].

Tumor-associated fibroblasts (TAFs) within the TME of pituitary adenomas play a crucial biological role. Cytokines released from TAFs can impact both tumoral and non-tumoral cells, including macrophages. This interaction may enhance tumor invasiveness and influence angiogenesis and epithelial-to-mesenchymal pathways in PAs. IL-6 and CCL2 have been identified as key factors in this process. Intriguingly, the somatostatin analogue pasireotide has been found to inhibit the secretions of TAFs, suggesting a potential anti-tumoral effect of somatostatin analogues (SSAs) by directly targeting TAFs and thus altering the TME in PAs [60] (Figure 2).

Recent studies, both preclinical and clinical, have underscored the importance of anti-VEGF therapy in treating pituitary tumors [61]. In murine models, anti-VEGF agents have demonstrated significant effectiveness when used as single agents. Bevacizumab, a recombinant humanized monoclonal antibody that targets VEGF, has gained recognition as the first approved agent in this class. Its emergence marks a notable development in the therapeutic options available for pituitary tumors.

## 7. Biomarkers: The Future of Diagnosis and Treatment

### 7.1. The Importance of Biomarkers in Oncology

In oncology, biomarkers play a multifaceted and essential role, spanning risk assessment, screening, differential diagnosis, prognosis, the prediction of response to treatment, and monitoring cancer progression. The rapid advancement in technology has spurred extensive research into a wide array of potential biomarkers, leading to significant developments in this field [62]. These biomarkers are varied, encompassing different biochemical forms such as nucleic acids, proteins, carbohydrates, lipids, small metabolic products, cytogenetic and cytokinetic indicators, and entire tumor cells in bodily fluids [63].

The search for biomarkers that can predict response to immunotherapy is an active and growing area of research. A deeper understanding of immune system dynamics is anticipated to enhance the efficacy of immunotherapy, allowing for treatments to be tailored to the specific needs of individual patients. Central to this effort is the development and improvement of analytical platforms and assays for accurate and consistent biomarker quantification both within and between individuals [64].

In cancer biology, cancer stem cells (CSCs) are a unique subset of tumor cells capable of initiating tumors and driving recurrence. These cells can originate from either differentiated cells or adult tissue stem cells. Future therapeutic strategies are being shaped to focus on targeted immune-mediated interventions, including chimeric antigen receptor-based methods, to eliminate cells resistant to chemotherapy and CSCs. This approach is paving the way for personalized therapeutic strategies [65].

In miRNA research, a plethora of potential cancer biomarkers have been identified, contributing to advances in cancer diagnosis and prognosis. This has led to new approaches in cancer screening. Simultaneously, various miRNA-based therapies are being investigated for different types of cancer, with combined treatment methods and the use of additional miRNAs showing potential in improving clinical outcomes for cancer patients [66].

Advances in genomics, proteomics, and molecular pathology have resulted in the identification of numerous potential biomarkers with clinical relevance. Their application in cancer staging and in customizing treatments at diagnosis could significantly enhance patient management. Understanding how to optimally integrate these biomarkers into clinical practice is crucial for translating their theoretical benefits into real-world clinical improvements.

### 7.2. Exploring Potential Biomarkers Specific to Intracranial Tumors and Pituitary Adenomas

Several biomarkers have been identified that possess predictive value for managing pituitary tumors, particularly concerning their clinical and radiological characteristics.

The cell cycle is regulated by key components such as cyclins, cyclin-dependent kinases (CDKs), and their inhibitors (CDKIs). There are two main families of CDKIs: the INK family (including INK4a/p16, INK4b/p15, INK4c/p18, and INK4d/p19) and the WAF/KIP family (comprising WAF1/p21, KIP1/p27, and KIP2/p57). The progression through the cell cycle is largely controlled by fluctuations in the levels of cyclins and CDKIs, which are regulated via programmed degradation by the ubiquitin–proteasome system [67].

In adrenocorticotrophic hormone-secreting pituitary tumors, the roles of matrix metalloproteinase-9 (MMP-9), pituitary tumor-transforming gene (PTTG), and high mobility group A 2 (HMGA2) in tumor development are well established. However, their relationships with tumor recurrence following transsphenoidal adenomectomy remain unclear. Notably, ACTH-secreting pituitary tumors with increased levels of MMP-9 have been associated with higher recurrence rates and shorter recurrence-free survival periods [68].

Vascular endothelial growth factor plays a vital role in angiogenesis, a process critical in both developmental stages and pathological conditions in pituitary tumors. A significant amount of preclinical and clinical research has highlighted the importance of anti-VEGF therapy in treating pituitary tumors [51,69]. Additionally, the expression of Galectin-3 (LGALS3) has been recognized as a predictive factor for the aggressive behavior of PRL and ACTH-functioning pituitary adenomas, with its expression levels correlating with those of RUNX1 [70].

### 7.3. Role of Biomarkers in Early Diagnosis and Targeted Therapies

Pituitary adenomas exhibit varied behaviors, and predicting their potential for aggressive or malignant transformation is complex. Biomarkers are increasingly recognized as vital in this context, including a range of factors such as chromosomal changes, microRNAs, markers of cellular proliferation, oncogenes, tumor suppressor genes, growth factors and their receptors, and elements related to angiogenesis or cell adhesion [71].

Recent research has identified three RNA subtypes, messenger-RNA, long non-coding RNA, and micro-RNA, that show correlations with PitNETs. However, their exploration in liquid biopsies remains limited, with only a few studies conducted so far. Similar to circulating tumor DNA, cell-free RNA is predominantly found within extracellular vesicles, particularly exosomes, rendering it a valuable component for liquid biopsy research [72].

Investigations into the use of bevacizumab for treating aggressive pituitary adenomas, particularly after standard therapies have been exhausted, show potential efficacy. However, factors such as the advanced disease stage of the patients, the brief treatment duration, and the limited clinical benefit observed in a single case make it challenging to conclusively assess its overall effectiveness [73].

With the recent success of immune checkpoint inhibitors in various solid cancers, interest in the potential of immunotherapy for aggressive, refractory pituitary tumors is growing. Preliminary findings suggest that pembrolizumab might be effective in treating certain subtypes of pituitary carcinoma, especially corticotroph tumors. These results encourage further investigations to refine treatment approaches and identify biomarkers for predicting responses to immunotherapy in these tumors [15].

## 8. Old and Emerging Targets for Therapy: A Glimpse into the Future

### 8.1. Historical Treatment Options: A Static Landscape

#### 8.1.1. Analysis of Traditional Treatments over the Past 20 Years

In the past two decades, there has been significant progress in the treatment approaches for intracranial tumors. Radiation therapy, traditionally key in treating malignant and aggressive intracranial tumors, has proven its effectiveness by extending patient survival rates and improving tumor control. Recent advances in surgical and radiotherapeutic techniques have re-emphasized the importance of radiation, not just as a primary but also as a secondary treatment option for benign tumors [74].

Radiotherapy (RT) is a core element in the management of malignant tumors and is increasingly being applied to benign conditions. The effectiveness of low to intermediate doses of RT has been extensively studied. However, the application of post-operative radiotherapy (PORT) continues to be debated, in spite of numerous trials and meta-analyses conducted over many years [75].

The execution of large-scale, multicenter therapeutic trials remains a challenge. Future research is expected to increasingly depend on data from translational clinical trials, particularly those using canine intracranial tumor models. Continuous data collection and analysis regarding the natural biology and clinical progression of specific tumor types and grades are crucial for evaluating therapeutic methods. This requires a minimum histologic diagnosis for publication. Furthermore, with the evolution of molecular-based therapies, the earlier mentioned advancements in diagnostic classification are anticipated to become more significant [76].

#### 8.1.2. The Exceptions: Pituitary Adenomas and the Alternative Treatments

Pituitary adenomas constitute around 15% of all brain tumors, and their detection is increasingly common, largely due to the widespread use of magnetic resonance imaging. Surgical intervention is the primary treatment for most of these tumors. The effectiveness of dopaminergic agonists and somatostatin receptor ligands (SRLs) in treating specific types of pituitary adenomas, notably prolactinomas and growth hormone excess, is well established. Over the past decade, there has been an increase in the application of new multi-receptor binding SRLs for acromegaly and Cushing’s disease treatment [77].

Functional pituitary adenomas, which can cause morbidity through hormone hypersecretion, are often effectively managed with medical therapies aimed at inhibiting pituitary hormone secretion or the response of target organs. These medical interventions are non-invasive and mitigate the anatomical and potentially permanent risks associated with surgical and radiation treatments. However, aside from prolactinomas, typically medical therapies are not curative and are unlikely to fully eliminate the adenoma. The main objective of these therapies is to attain biochemical control, and treatment risks vary based on the specific medication. Considering that these therapies might be prolonged or indefinite, understanding patient tolerability is essential [14].

Radiotherapy has been proven effective in managing pituitary adenomas, achieving 97.9% radiological control and 93.6% biochemical control over a median six-year follow-up period post radiotherapy [78]. Techniques like intensity-modulated radiation therapy, stereotactic radiosurgery, and proton beam radiation therapy, which focus energy beams precisely on the tumor to minimize damage to adjacent healthy tissues, are employed [79].

In the field of minimally invasive surgery, the endoscopic transsphenoidal approach is highly efficient for removing pituitary tumors. This procedure can be conducted using either a one-hand or two-hand technique. The one-hand approach, involving access through a single nostril, is generally more straightforward for surgeons but may restrict instrument movement. The two-hand technique, utilizing a one-and-a-half nostril approach, offers more refined manipulation. Notably, the two-hand/one-and-a-half nostril technique in single-surgeon endoscopic endonasal transsphenoidal surgery has been shown to be safe and effective for excising large pituitary tumors [80,81].

### 8.2. Overview of the Current State of Targeted Therapies

Pituitary adenomas are linked with several related disorders, including prolactinoma, acromegaly, Cushing’s disease, and non-functioning pituitary adenoma. Treatment often requires a combination of surgery, medical therapies (such as dopamine agonists or somatostatin receptor ligands), and radiotherapy, due to their significant impact on patient mortality, morbidity, and quality of life [82,83]. Transsphenoidal surgery is usually the first-line treatment for pituitary tumors, except for prolactinomas. However, the tumor’s local invasiveness may complicate surgical resection, occasionally necessitating initial medical therapy [84].

Prolactinomas, the most common type of secretory pituitary tumors, can cause symptoms from prolactin oversecretion, localized mass effects, or both [85]. Traditionally, dopamine agonists have been the primary treatment, with cabergoline effectively normalizing prolactin levels in most patients and inducing tumor shrinkage in many. The surgical resection of microprolactinomas and encapsulated macroprolactinomas can achieve remission rates similar to cabergoline treatment [86].

Acromegaly, a chronic disease caused mainly by growth hormone-secreting PitNETs, can lead to severe health issues and increased mortality if not properly managed. Transsphenoidal surgery is preferred for GH-secreting PitNETs, with medical therapy, especially long-acting somatostatin analogs, used when surgery is not possible or incomplete, achieving control in about half of the patients [87].

Cushing’s disease, caused by an ACTH-secreting pituitary tumor, is the most common form of Cushing’s syndrome and requires prompt diagnosis and treatment to improve outcomes [88]. Surgery is the initial therapy, but when it is not curative, medical therapy becomes a significant secondary option. New drugs and formulations are being investigated for their efficacy and safety in treating Cushing’s disease [89]. Recent medical advancements include using cabergoline or pasireotide to target dopamine and somatostatin receptors on corticotroph adenomas, effectively controlling cortisol production in some patients [90].

Non-functioning pituitary adenomas, benign tumors that do not cause hormonal hypersecretion, are primarily treated with surgery, but recurrence rates are high [91]. Post surgery, dopamine agonists have been explored to prevent recurrence, showing some initial promise but diminishing effectiveness over time. The role of medical therapy in preventing NFPA regrowth remains limited [92].

In managing pituitary adenomas, including prolactinoma, acromegaly, Cushing’s disease, and non-functioning pituitary adenoma, transsphenoidal surgery is typically the primary treatment, except for prolactinomas. Sometimes, the tumor’s invasiveness may necessitate initial medical intervention.

Surgical resection maintains its status as a core strategy in managing brain tumors (BM), with its effectiveness in BM surgery being well established. A critical aspect of surgical intervention is the method employed, wherein en bloc resection, the removal of the tumor in a single piece, has been shown to offer better local control compared to piecemeal tumor resection. This approach also involves the minimal removal of adjacent normal brain tissue [93]. En bloc resection is notable for its potential to decrease intraoperative tumor spread relative to piecemeal resection. Patel et al. observed that en bloc resection does not correlate with an increase in complications or adverse functional outcomes [94]. Additionally, studies by Suki et al. have supported the efficacy of en bloc resection in reducing intraoperative tumor dissemination [95,96]. However, en bloc resection may not always be viable, particularly in cases of larger or cystic tumors. During such surgeries, it is important to minimize the leakage of internal contents and the spread of tumor fragments. After resection, an exhaustive inspection of the cavity is crucial to confirm the complete removal of the tumor, especially following piecemeal resection.

Most pituitary adenomas can be effectively managed with current medical treatments, surgical interventions, and, in some cases, radiotherapy. However, gonadotroph adenomas are particularly challenging due to the lack of effective medical treatments. Additionally, the management of aggressive pituitary adenomas and pituitary carcinomas presents significant clinical challenges [97].

For prolactinomas, the use of dopamine agonists has been a primary treatment strategy since the 1970s. The initial use of bromocriptine later shifted to cabergoline, which has been effective in regulating prolactin levels in up to 85% of patients and reducing tumor size in about 80% of cases. The surgical removal of both microprolactinomas and larger macroprolactinomas, when performed by neurosurgeons specialized in pituitary disorders, has achieved remission rates comparable to those obtained with cabergoline therapy. For particularly aggressive prolactinomas and metastasized PitNETs, a comprehensive treatment approach is recommended. This includes higher doses of cabergoline, surgical intervention, radiation therapy (preferably stereotactic radiosurgery when possible), and temozolomide administration. While DAs remain effective for most prolactinoma cases, the success rates of transsphenoidal surgical methods have significantly improved recently, especially with the expertise of specialized surgeons [98].

In cases where hypercortisolism persists or recurs after transsphenoidal surgery, or when surgery is not an option or is declined by the patient, medical therapy becomes crucial. This treatment primarily focuses on three areas: inhibiting the production of ACTH by corticotroph tumors, suppressing steroidogenesis in the adrenal glands, and blocking the action of glucocorticoid receptors. Pituitary-directed medications include pasireotide and cabergoline. Pasireotide has demonstrated effectiveness in reducing cortisol levels in patients with Cushing’s disease, indicating its utility in treating corticotropin-secreting pituitary adenomas [99,100]. Cabergoline, when used as a sole treatment, can be effective over the long term for select Cushing’s disease patients, although it requires ongoing monitoring for dose adjustments [101]. Steroidogenesis inhibitors currently used include ketoconazole, metyrapone, mitotane, and etomidate. The management of Cushing’s syndrome, particularly Cushing’s disease, remains complex, with adrenal steroidogenesis inhibitors playing a key role in controlling hypercortisolism at various stages, both before and after surgery.

In the case of growth hormone pituitary adenoma (GHPA), new pharmacological approaches have been employed. The role of p300, a histone acetyltransferase (HAT) coactivator implicated in PA tumorigenesis and progression, and its catalytic inhibitors in GHPA is an active area of research. Recent findings indicate that using the selective inhibitor A-485 to target HAT p300 may be a promising treatment strategy for GHPA [102].

For thyrotropin-secreting pituitary neuroendocrine tumors, employing high-sensitivity TSH detection methods is crucial for early diagnosis and effective treatment. Advances in MRI techniques have enhanced the non-invasive detection of smaller PitNETs. Treatment options for TSH PitNETs include surgery, pharmacological treatments, and radiotherapy [103].

### 8.3. Potential Novel Targets

Innovative targeted therapies for pituitary tumor treatment are focusing on inhibitors of the mammalian target of rapamycin (mTOR), tyrosine kinase, and vascular endothelial growth factor. Advances in the molecular characterization of endocrine tumors through genomic, epigenomic, and transcriptomic analyses have provided a more in-depth understanding of their pathogenesis and molecular pathology. This has led to an increased application of molecular targeted therapies (MTTs) in patient care [104]. The mTOR protein kinase is vital for regulating various cellular processes, including metabolism, catabolism, immune responses, autophagy, survival, proliferation, and migration, thus maintaining cellular homeostasis. Recent studies have shown that the constitutive activation of the mTOR pathway, due to genetic changes like mutations, amplifications, or deletions in mTOR itself, its complexes (mTORC1 and mTORC2), or its upstream targets, plays a role in aging, neurological disorders, and various human cancers. Progress in targeting mTOR signaling could significantly improve anti-cancer treatments and benefit patients in clinical settings [105]. 

Immune checkpoint treatment represents a very promising therapeutic avenue for neoplasms [106,107,108]. This therapeutic avenue is also a very hopeful option for pituitary adenomas. There is a well-established correlation between mismatch repair deficiency (MMRd) and the heightened responsiveness to anti-PD-1 therapies in various malignancies. This association can be effectively identified through the immunohistochemical analysis of markers such as MLH1, MSH2, MHS6, and PMS2. An instance of accelerated disease progression was reported in the literature following the administration of pembrolizumab in a patient diagnosed with an MMRd pituitary ACTH-secreting adenoma, showcasing the limited efficacy observed with pembrolizumab in the treatment of an ACTH-secreting pituitary adenoma characterized by mismatch repair deficiency, potentially attributable to elevated cortisol levels in the patient [50]. Furthermore, the expression of PD-L1 is prevalent in functional pituitary adenomas, concomitant with an association with aggressive clinical manifestations in these adenomas [109]. The therapeutic blockade of the CTLA-4 and PD-1 pathways plays a pivotal role in modulating immune response against tumors. This blockade enables the activation of T cell stimulatory signaling pathways, subsequently augmenting antitumor T cell cytotoxicity, promoting the production of proinflammatory cytokines, stimulating T cell proliferation, and facilitating tumor cell destruction. The concurrent inhibition of the CTLA-4 and PD-1/PD-L1 pathways, owing to their distinct and nonredundant coinhibitory functions, has been observed to yield superior clinical remission rates [110,111,112,113]. However, this combination therapy is associated with an increased incidence of immune-related toxicities. While typically well tolerated, the combined therapeutic approach elevates the likelihood of encountering high-grade immune-related adverse events (irAEs). Within the clinical treatment paradigm, irAEs precipitated by ICIs exhibit a toxicity profile that is often more pronounced than that of conventional chemotherapy, encompassing a range of organ systems. Pertaining to the gastrointestinal system, afflictions such as colitis, hepatitis, cholangitis, and gastritis frequently manifest as common irAEs. Conversely, in the hematologic domain, these adverse events are typically characterized by autoimmune hemolytic anemia, immune thrombocytopenia, and aplastic anemia [114,115].

Regarding tyrosine kinases, both preclinical and clinical research has underscored the importance of the EGFR pathway in corticotroph and lactotroph adenomas. While further investigation is needed, current data suggest that targeting the ErbB pathway may be an effective therapeutic approach for patients with aggressive pituitary tumors.

The epigenetic novel markers as well as targeted proteins in current emerging therapies mentioned throughout our study are comprehensively summarized for a broader view of those therapeutic possibilities (Table 2).

#### 8.3.1. Advances in the Imaging of Pituitary Tumors

Magnetic resonance imaging is the primary imaging method for diagnosing pituitary tumors. In the MRI scans of nonfunctioning pituitary adenomas, asymmetry is commonly observed. Typically, the tumor tends to extend more freely laterally on the side adjacent to the cavernous sinus. In contrast, lateral extension is less pronounced on the side where the normal pituitary gland is located. A notable feature often seen on the side of the normal pituitary gland is the immobile position of the internal carotid artery (ICA), creating an ‘ICA notch.’ This feature can be a useful preoperative indicator to identify the side of the normal pituitary gland [116].

Using alternative MRI sequences can be crucial in identifying specific characteristics of the tumor, such as its consistency, the presence of apoplexy, or invasion into adjacent structures. Additionally, various radiotracers have been investigated for the molecular imaging of pituitary tumors, yielding mixed results in terms of efficacy [117].

Positron emission tomography with 11C-methionine (MET-PET) has been suggested as a sensitive technique that enhances MRI for detecting residual or recurrent pituitary adenomas. Due to its high sensitivity, MET-PET is poised to become an important tool in the effective management of these tumors [118].

#### 8.3.2. Single-Cell RNA Sequencing

A comprehensive study employed high-precision scRNA-seq on 2679 individual cells obtained from 23 surgically removed samples of various PitNET subtypes from 21 patients. This research led to the discovery of several new tumor-related genes, including AMIGO2, ZFP36, BTG1, and DLG5. It was found that tumors expressing multiple hormone genes showed minimal transcriptomic variability. Additionally, single-cell multi-omics analyses indicated that these tumors had a relatively uniform genomic pattern, with only slight differences in copy number variations [42]. This study highlights the potential of scRNAseq technologies to significantly advance the understanding of the development and functioning of diverse cell populations within the pituitary gland throughout a person’s lifetime [119].

## 9. Conclusions and Future Prospects

Our comprehensive review of pituitary adenomas has navigated the complex terrain of these challenging brain tumors, illuminating various facets that contribute to our understanding of these conditions.

We began by examining the evolution of cancer treatment modalities over the past two decades, focusing specifically on the unique challenges and alternative therapeutic strategies associated with pituitary adenomas. An in-depth overview of these diseases followed, exploring how current and emerging treatments could impact their complex biological mechanisms. Critical areas like epigenetics, transcriptomics, and immunology were explored in depth, providing insights into the intricate nature of pituitary tumors. For each subtype, the potential of biomarker-targeted therapies was examined, offering perspectives on future directions in understanding and managing these complex conditions.

Cancer is characterized by the uncontrolled and abnormal proliferation of cells. A key gene in cell cycle regulation, acting as a tumor suppressor, is p53, often called the “guardian of the genome”, and encoded by the TP53 gene. P53 can be activated by various stimuli, including DNA damage, heat shock, hypoxia, and oncogene overexpression. As a regulatory protein, p53 orchestrates a range of biological responses and plays a vital role in maintaining genetic stability by preventing genomic mutations [120].

Traditional cancer treatment has primarily depended on surgery, radiotherapy, and chemotherapy. However, recent advances in cancer research have led to an interest in integrating these traditional methods with newer approaches. The growing realization that cancer progression involves not just changes in cancerous epithelial cells but also in the tumor microenvironment has spurred the development of new combination therapies. This comprehensive view of cancer development and progression is leading to innovative treatment strategies [121].

To conclude this review, it is crucial to look forward to the trajectory of brain tumor research. We have highlighted significant advancements and innovations from current scientific literature, which promise to transform treatment approaches and potentially address the challenges posed by these relentless diseases.

## Figures and Tables

**Figure 1 biomedicines-12-00023-f001:**
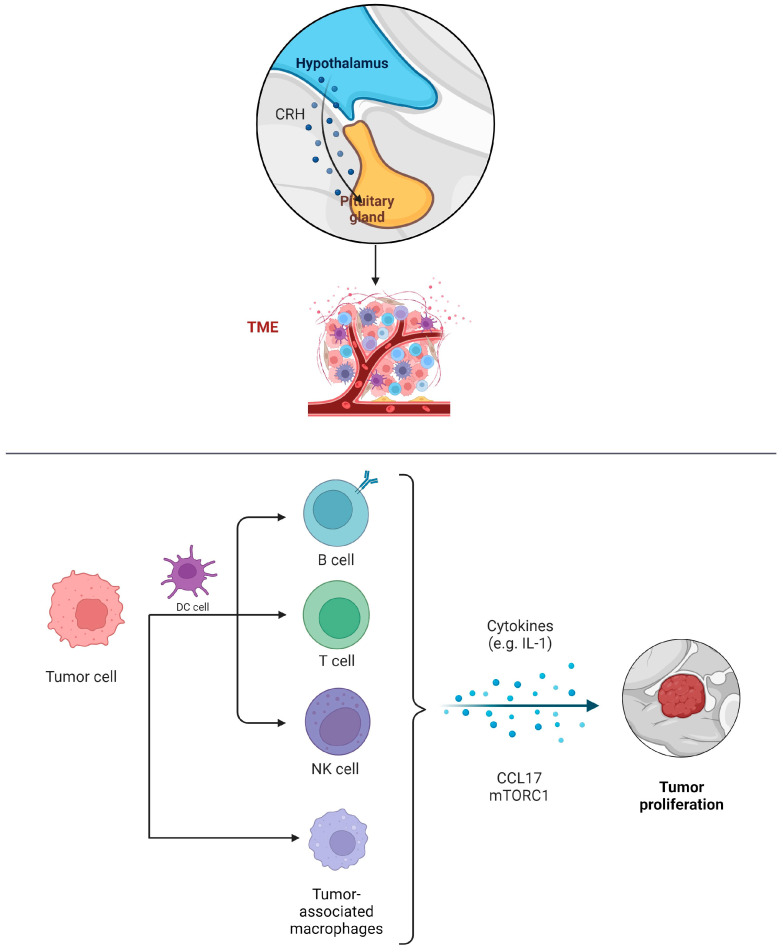
Infiltration of immune cells into PAs and their potential in the invasion, migration, and proliferation of PAs.

**Figure 2 biomedicines-12-00023-f002:**
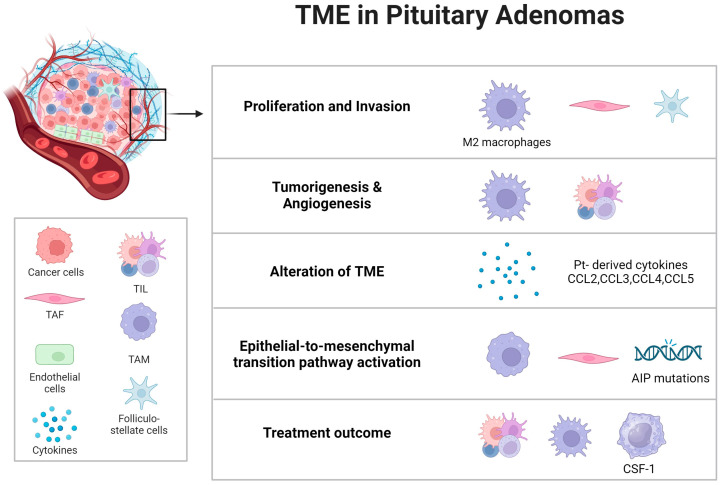
Microenvironmental influence on pituitary tumor dynamics. An examination of the pituitary tumor microenvironment’s composition and its profound influence on tumor behavior, with specific emphasis on critical tumorigenic processes such as tumor cell proliferation, invasion, tumor initiation, angiogenesis, the activation of the epithelial-to-mesenchymal transition pathway, the modulation of the microenvironment, and responsiveness to therapeutic interventions. Abbreviations: Pt = pituitary tumors; TME = tumor microenvironment; TAM = tumor-associated macrophages; TAF = tumor-associated fibroblasts; TIL = tumor-infiltrating lymphocytes; AIP = aryl hydrocarbon receptor-interacting protein; CSF-1 = colony-stimulating factor 1.

**Table 1 biomedicines-12-00023-t001:** Possible immune checkpoints of significance in pituitary tumors.

Immune Modulators	Function in Pituitary Carcinomas (pc)	Suggestions	References
PD-1/PD-L1	Cushing’s disease ACTH-pc Prolactin-pc	For high PD-L1 pc	[48,49,50,51]
CTLA-4	ACTH-pc Prolactin-pc	Therapeutic combination of CTLA-4 and PD-1/PD-L1 inhibitors	[49,51]

PD-1/PD-L1: programmed cell death protein 1/programmed cell death ligand 1; CTLA-4: cytotoxic T-lymphocyte antigen, 4.

**Table 2 biomedicines-12-00023-t002:** Epigenetic markers/target proteins.

Epigenetic Marker/Target Protein	Role/Impact	Study Reference
DNA methylation (on TSGs)	Selectively affects TSG expression and tumor characteristics in PitNETs	[19]
Histone modifications (p53 expression)	Linked to increased p53 expression and longer progression-free survival in pituitary tumors	[21]
Sirtuins	Higher expression in growth hormone-expressing adenomas, inverse correlation with tumor size	[21]
Citrullinating enzymes	Proposed as an early marker in prolactinomas pathogenesis	[21]
DNMTs and HDACs	Targeted by epigenetic therapies to reactivate silenced genes, enhancing sensitivity to conventional treatments	[22]
miRNAs (various)	Implicated in pituitary tumorigenesis, acting as oncosuppressors or oncogenes	[23,24,25]
HDAC2 and HDAC3	Higher expression in clinically non-functioning pituitary adenomas, potential therapeutic targets	[26]
miR-7a2 and its target gene Raf1	Involved in lactotroph cell development and prolactin synthesis, potential therapeutic target in hyperprolactinemia	[23]
BCAT1 (in PRL-PAs)	Associated with increased prolactin production, drug resistance, and proliferation	[34]
NR5A1, TBX19, POU1F1 (transcription factors)	Guide terminal differentiation in pituitary adenomas, categorizing them into molecular clusters	[35]
USP8 (in corticotrophinomas)	Mutations lead to the prolonged activation of EGFR, stimulating ACTH synthesis	[38,39]
DRD2 and NGFR (in prolactinomas)	Associated with response to dopamine agonists (DAs)	[41]
AMIGO2, ZFP36, BTG1, DLG5 (in gonadotroph tumors)	Identified as novel tumor-related genes	[42]
HSPB1	Associated with reduced overall survival in tumors, potential therapeutic target in invasive pituitary adenoma	[46]
Immune Checkpoint Blockers	Explored as a new therapeutic option for aggressive pituitary adenomas/PitNETs and pituitary carcinomas	[47,59]
IL-6, TNF-α	Play significant roles in the development and progression of invasive pituitary adenoma	[56,57]
VEGF/VEGFR Signaling	Associated with an immunosuppressive microenvironment in non-functioning PitNETs with CS invasion, potential target for therapy	[58]
Tumor-Associated Fibroblasts (TAFs)	Impact tumor invasiveness, angiogenesis, and epithelial-to-mesenchymal pathways in PAs, target of SSAs	[60]
mTOR Inhibitors	Target the constitutive activation of the mTOR pathway in various human cancers, including pituitary tumors	[104,105]
Mismatch Repair Deficiency (MMRd) and PD-L1 Expression	Correlated with responsiveness to anti-PD-1 therapies; PD-L1 prevalent in functional pituitary adenomas	[50,109]
Tyrosine Kinase Inhibitors (targeting EGFR)	Potential therapeutic approach for aggressive corticotroph and lactotroph adenomas	[78,104]

## Data Availability

All data are available online on libraries such as PubMed.

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
