# Peer review of "From Genes to Therapy: Pituitary Adenomas in the Era of Precision Medicine"

_biomedicines, 2023, doi:10.3390/biomedicines12010023_

Round 1
Reviewer 1 Report
Comments and Suggestions for Authors
Dear authors;
I was really enjoying your’ manuscript. All presented sections are readable, sophisticated but understandable. I have just one suggestion (to better visualisation of study), could you prepare sum-up in the table: epi-markers, target proteins for tumors/feature and current treatment target.
Thank you for your manuscript and whish you fruitful feature work.
Author Response
Dear Reviewer,
We are deeply grateful for the insightful feedback provided on our manuscript. Your constructive comments have been instrumental in enhancing the quality and clarity of our work.
We sincerely appreciate your positive remarks regarding the readability and sophistication of our manuscript. In response to your valuable suggestion, we are pleased to incorporate a summary table that outlines the epigenetic markers, target proteins for tumors/features, and current treatment targets. This table will be designed to provide a clear, comprehensive visualization of the study's findings, thereby enriching the manuscript's informational value. Thank you for your encouragement and well-wishes for our future work.
Thank you for your thorough review and valuable suggestions. Your guidance is instrumental in refining our manuscript and advancing our research.
Sincerely,
The collective of authors
Reviewer 2 Report
Comments and Suggestions for Authors
The literature review done by Toader et al. it is very precise and complete regarding the etiopathogenesis, biological markers and possible treatments of pituitary adenomas. However, in my opinion, some changes are necessary.
1. In the Introduction it would be necessary to reduce paragraphs 1.2, 2.1 and 2.2, since they contain information that is not useful to the main topic, which concerns pituitary adenomas, and distracts the reader. Therefore I would recommend removing these paragraphs by inserting some more relevant information in the following paragraphs (of chapter 9) dedicated to the therapy of the aforementioned tumors. Obviously, the abstract regarding this part (lines 20-24) must be modified accordingly.
2. The same can be said about paragraph 3.3, which is partly taken up in the following paragraph 9.1. Also in this case, I would suggest unifying the two paragraphs, reporting the information regarding the classic therapy of pituitary adenomas only in paragraph 9.1
3. There are some typographical errors to correct, such as line 453 (contributes) or line 592 (of). Furthermore on page 14, line 656, there is a sentence that is out of context (The role of technology and innovation in uncovering these targets) that needs to be removed.
4. Finally, the Authors forgot to include, at the end of the manuscript (lines 714-717), information about the Authors' Contribution, funding, data availability statement and conflict of interest.
Author Response
Dear Reviewer,
We are deeply grateful for the insightful feedback provided on our manuscript. Your constructive comments have been instrumental in enhancing the quality and clarity of our work.
- Introduction Revision: We acknowledge your insightful observation regarding the superfluous information in paragraphs 1.2, 2.1, and 2.2 of the Introduction. We agree that streamlining these sections will enhance focus and relevance to the main topic of pituitary adenomas. Accordingly, we have modified these paragraphs and added more insights into CTLA-4 PD-L1 therapy in section 9. The abstract will also be revised to reflect these changes.
- Paragraphs 3.3 and 9.1 Consolidation: Your recommendation to unify paragraphs 3.3 and 9.1 is well-received. We will merge these sections to present a cohesive and comprehensive account of the classic therapy for pituitary adenomas, eliminating redundancy and enhancing the manuscript's coherence.
- Correction of Typographical Errors: We are grateful for your attention to detail in identifying typographical errors. The specified errors on line 453, line 592, and the out-of-context sentence on page 14, line 656, will be promptly corrected. We understand the importance of precision in academic writing and appreciate your assistance in improving our manuscript.
- Inclusion of Essential Information: Lastly, we apologize for the oversight regarding the omission of the Authors' Contribution, funding, data availability statement, and conflict of interest at the end of the manuscript.
Thank you for your thorough review and valuable suggestions. Your guidance is instrumental in refining our manuscript and advancing our research.
Sincerely,
The collective of authors
Round 2
Reviewer 2 Report
Comments and Suggestions for Authors
The authors have reworked the review but in my opinion some doubts and errors remain.
Regarding the former, the main one is that by reading the introductory part several times, I was unable to understand the need to include chapter 2 to deal with traditional therapies and alternative treatments regarding pituitary adenomas. My advice is to delete chapter 2 and start after the introduction with chapter 3. Chapter 2, however, can be inserted into chapter 9 (paragraph 9.1, which obviously needs to be reworked to include information included in chapter 2), modifying the title of the same in "Old and emerging targets for therapy: a glimpse into the future".
About the errors, they are as follows:
1. magnetic resonance is initially abbreviated as MR, but later it is MRI (I stands for imaging): the authors have to choose which acronym to use
2. page 11, line 490: of is written using zero and not O
3. The titles of paragraphs 9.2.1. and 9.2.2 must be written in italic
4. Table 2 has been inserted, but it is not cited in the text and in any case should be positioned elsewhere in the text
Author Response
Dear reviewer,
We appreciate your time in reviewing our manuscript, all indications were respected accordingly
Chapter 2 was deleted and positioned in chapter 9, renaming this part of our manuscript
- We decide to use MRI in the whole manuscript
- Term was revised
- Titles are in italic format now
- Table 2 is now cited in the text and positioned after potential novel targets part (now chapter 8.3), even though our table is named "epigenetic markers/target proteins" we could not insert Table 2 in the part discussing epigenetics of pituitary adenoma (now chapter 3) because this final table summarized all of our research and had to be arranged at the end of our manuscript
Thank you for your indications, all recommendations were undergone
Sincerely,
The collective of authors